# GPR183 Regulates 7α,25-Dihydroxycholesterol-Induced Oxiapoptophagy in L929 Mouse Fibroblast Cell

**DOI:** 10.3390/molecules27154798

**Published:** 2022-07-27

**Authors:** Jae-Sung Kim, HyangI Lim, Jeong-Yeon Seo, Kyeong-Rok Kang, Sun-Kyoung Yu, Chun Sung Kim, Do Kyung Kim, Heung-Joong Kim, Yo-Seob Seo, Gyeong-Je Lee, Jae-Seek You, Ji-Su Oh

**Affiliations:** 1Institute of Dental Science, School of Dentistry, Chosun University, Gwangju 61452, Korea; js_kim@chosun.ac.kr (J.-S.K.); qjqtjdgod@naver.com (H.L.); sj23850126@hanmail.net (J.-Y.S.); kkr@chosun.ac.kr (K.-R.K.); sky@chosun.ac.kr (S.-K.Y.); cskim2@chosun.ac.kr (C.S.K.); kdk@chosun.ac.kr (D.K.K.); hjbkim@chosun.ac.kr (H.-J.K.); 2Department of Oral and Maxillofacial Radiology, School of Dentistry, Chosun University, Gwangju 61452, Korea; moresys@chosun.ac.kr; 3Department of Prosthodontics, School of Dentistry, Chosun University, Gwangju 61452, Korea; lkj1998@chosun.ac.kr; 4Department of Oral and Maxillofacial Surgery, School of Dentistry, Chosun University, Gwangju 61452, Korea; applit375@chosun.ac.kr

**Keywords:** cholesterol, oxysterol, 7α,25-dihydroxycholesterol, oxiapoptophagy, apoptosis, oxidative stress, autophagy

## Abstract

7α,25-dihydroxycholesterol (7α,25-DHC) is an oxysterol synthesized from 25-hydroxycholesterol by cytochrome P450 family 7 subfamily B member 1 (CYP7B1) and is a monooxygenase (oxysterol-7α-hydroxylase) expressed under inflammatory conditions in various cell types. In this study, we verified that 7α,25-DHC-induced oxiapoptophagy is mediated by apoptosis, oxidative stress, and autophagy in L929 mouse fibroblasts. MTT assays and live/dead cell staining revealed that cytotoxicity was increased by 7α,25-DHC in L929 cells. Consequentially, cells with condensed chromatin and altered morphology were enhanced in L929 cells incubated with 7α,25-DHC for 48 h. Furthermore, apoptotic population was increased by 7α,25-DHC exposure through the cascade activation of caspase-9, caspase-3, and poly (ADP-ribose) polymerase in the intrinsic pathway of apoptosis in these cells. 7α,25-DHC upregulated reactive oxygen species (ROS) in L929 cells. Expression of autophagy biomarkers, including beclin-1 and LC3, was significantly increased by 7α,25-DHC treatment in L929 cells. 7α,25-DHC inhibits the phosphorylation of Akt associated with autophagy and increases p53 expression in L929 cells. In addition, inhibition of G-protein-coupled receptor 183 (GPR183), a receptor of 7α,25-DHC, using GPR183 specific antagonist NIBR189 suppressed 7α,25-DHC-induced apoptosis, ROS production, and autophagy in L929 cells. Collectively, GPR183 regulates 7α,25-DHC-induced oxiapoptophagy in L929 cells.

## 1. Introduction

Oxysterols are cholesterol derivatives with a steroid backbone composed of seven carbon molecules with methylheptyl side chains [1]. They are synthesized from cholesterol by enzymatic and/or non-enzymatic pathways through encounters with reactive oxygen species (ROS) during sterol metabolism [2,3]. Oxysterols are bioactive molecules that are closely associated with the regulation of cholesterol homeostasis, lipid metabolism, and intermediates in the synthesis of bile acid and steroid hormones, and they act as ligands for nuclear and G-protein-coupled receptors (GPCR) [3]. Although oxysterols are involved in various physiological processes including apoptosis [4], autophagy [5], and prenylation of proteins [6], their effects remain largely unknown [7]. Previous studies have reported that certain oxysterols are physiological risk factors associated with age-related disorders, including cardiovascular and neurodegenerative diseases, osteoarthritis, and tumorigenesis [8,9,10].

25-hydroxycholesterol (25-HC) is an oxysterol with a hydroxyl group at the 25th carbon of cholesterol that is mediated by cholesterol-25-hydroxylase, which is produced by ROS [11]. As shown in Figure 1, 7α,25-dihydroxycholesterol (7α,25-DHC) is an oxysterol formed from 25-HC by cytochrome P450 family 7 subfamily B member 1 (CYP7B1), also known as steroid 7α-hydroxylase [12]. In particular, 7α,25-DHC has been identified as a potential natural ligand for GPR183 (Epstein–Barr virus-induced gene 2) [13]. Recent studies have reported that the 7α,25-DHC-GPR183 axis is closely associated with immune regulation, including innate and adaptive immunity [13,14], regulation of neuropathic pain in the spinal cord [15], and regulation of cytokine release through modulation of proinflammatory signaling in astrocytes [16]. Furthermore, studies have also reported that 7α,25-DHC participates in chemokine-induced migration of B, T, and dendritic cells [13]. However, the physiological and biological roles of 7α,25-DHC are poorly understood compared with those of 25-HC. Although many studies reported that 25-HC, an oxysterol precursor of 7α,25-DHC, induced apoptosis in various types of cell including human aortic smooth muscle cells [17], human keratinocytes [18], human colorectal cancer DLD-1 cells [19], adult rat Leydig cells [20], mouse macrophages [21], and rat chondrocytes [10], the cytotoxic properties of 7α,25-DHC remain largely unknown. Furthermore, Bartlett et al. reported that the 7α,25-DHC receptor regulates interferons and autophagy in primary human monocytes during Mycobacterium tuberculosis infection [22].

Recently, cytotoxic dosages of oxysterols have been shown to induce cell death with complex apoptotic characteristics, oxidative stress, and autophagy, defined as oxiapoptophagy [23]. We hypothesized that 7α,25-DHC, a downstream oxysterol of 25-HC that is cytotoxic to various cell types, might induce apoptotic cells involved in oxidative stress and autophagy. Fibroblasts are the most common cells to compose the connective tissues through the formation of stroma in animals [24]. Hence, an in vitro cytotoxicity assay was developed to verify the toxicity of reagents or materials based on fibroblasts [25]. Especially, the L929 fibroblast cell line is specialized to use the toxicity assay [25].

Therefore, this study aimed to evaluate 7α,25-DHC-induced oxiapoptophagic cell death that is regulated by GPR183 in the L929 mouse fibroblast cell line.

## 2. Results

### 2.1. 7α,25-DHC Increased the Cytotoxicity of L929 Cells

The relative viabilities using 3-(4, 5-Dimethylthiazol-2-yl)-2,5-diphenyl tetrazolium bromide (MTT, Sigma-Aldrich, St. Louis, MO, USA) assay were 79.9 ± 5.2%, 60.7 ± 2.5%, 55.7 ± 6.3%, and 55.0 ± 3.8% in L929 cells incubated with 1, 5, 10, or 25 μg/mL 7α,25-DHC (Sigma-Aldrich) for 48 h, respectively, compared with control (100.2 ± 7.7%) (Figure 2a). Furthermore, 7α,25-DHC decreased live cells, which was stained with green fluorescence, in L929 cells (Figure 2b). As shown in Figure 2c, total cells were counted as 84.06 ± 7.9%, 64.04 ± 6.12%, 48.92 ± 5.65%, and 36.47 ± 10.33% by 1, 5, 10, or 25 μg/mL 7α,25-DHC in L929 cells, respectively, compared with control (100.51 ± 11.0%). However, of the total cells, the percentage of dead cells was 0.44 ± 0.8%, 3.121 ± 1.54%, 6.23 ± 2.89%, and 11.56 ± 3.83% in L929 cells incubated with 1, 5, 10, or 25 μg/mL 7α,25-DHC, respectively, compared with control (0.45 ± 0.8%). These data demonstrate that 7α,25-DHC induced death in L929 cells through increased cytotoxicity and decreased cell viability in a dose-dependent manner.

### 2.2. 7α,25-DHC Induced Apoptosis in L929 Cells

To examine whether 7α,25-DHC-induced cell death was related to apoptosis, L929 cells incubated with 1, 5, 10, or 25 μg/mL 7α,25-DHC for 48 h were stained with 4′,6-diamidino-2-phenylindole (DAPI; Sigma-Aldrich) (Figure 3a) and hematoxylin and eosin (H&E) (Figure 3b) to investigate chromatin condensation and morphological alterations, respectively. Cells that underwent apoptotic death, characterized by condensed chromatin and altered morphology as a cell shrinkage, were elevated in the L929 cells incubated with 7α,25-DHC. Furthermore, fluorescence-activated cell sorting (FACS) analysis using propidium iodide (PI) and annexin V labeling demonstrated that the population of dead cells, including apoptotic and necrotic cells, was 37.7%, 38.5%, 39.8%, and 46.2% in L929 cells incubated with 1, 5, 10, or 25 μg/mL 7α,25-DHC for 48 h, respectively, compared with control (8.8%) (Figure 4). These data indicate that 7α,25-DHC induces the apoptosis of L929 cells.

### 2.3. Cell Death Is Mediated by the Intrinsic Pathway of Apoptosis in L929 Cells Incubated with 7α,25-DHC

Western blot was performed using antibodies against pro- or anti-apoptotic molecules in L929 cells incubated with 1, 5, 10, or 25 μg/mL 7α,25-DHC for 48 h (Figure 5a,b). Anti-apoptotic molecules, such as B-cell lymphoma 2 (Bcl-2) and B-cell lymphoma-extra large (Bcl-xL), were decreased by 7α,25-DHC in L929 cells. In contrast, BH3 interacting-domain death agonist (Bid), a pro-apoptotic molecule belonging to the intrinsic apoptosis pathway, was decreased in L929 cells incubated with 7α,25-DHC. These data indicate that Bid was cleaved into a membrane-targeted death ligand, truncated Bid (tBid). Sequentially, the expression of Bcl-2-associated X protein (Bax), Bcl-2 homologous antagonist killer (Bak), and Bcl-2 associated agonist of cell death (Bad) induced the activation of caspase-9 in L929 cells incubated with 7α,25-DHC (Figure 5a). Furthermore, activated caspase-9 induced the cascade activation of caspase-3 and poly (ADP-ribose) polymerase (PARP) to induce cell death (Figure 5b,c). Our results demonstrate 7α,25-DHC-induced intrinsic apoptosis in L929 cells.

### 2.4. 7α,25-DHC Induces Inflammation through the Upregulation of ROS and Inflammatory Mediators in L929 Cells

ROS production was detected by 2,7-dichlorofluoroscin diacetate (DCFDA) staining and ELISA in L929 cells incubated with 1, 5, 10, and 25 μg/mL 7α,25-DHC for 48 h. ROS-positive cell numbers increased significantly after 7α,25-DHC exposure in L929 cells (Figure 6a). In addition, FACS analysis revealed that ROS intensities of ROS were 8.2%, 18.1%, 31.4%, and 43.7% by 1, 5, 10, or 25 μg/mL 7α,25-DHC in L929 cells, respectively, compared to control (5.3%). Furthermore, relative ROS intensities were 105 ± 5.25%, 113 ± 6.22%, 141.2 ± 4.94%, and 212 ± 16.96% by 1, 5, 10, or 25 μg/mL 7α,25-DHC in L929 cells, respectively, compared to control (100 ± 6.88%) (Figure 6b). As shown Figure 6c, the expression of inflammatory mediator COX-2 was significantly increased by 7α,25-DHC in L929 cells. Sequentially, productions of prostaglandin E_2_ (PGE_2_) downstream inflammatory mediator of COX-2 were measured by 617.7 ± 21.6, 692.3 ± 45.7, 902.03 ± 82.1, 1385.9 ± 72.1, and 1477.8 ± 82.8 pg/mL in L929 cells treated with 0, 1, 5, 10, and 25 μg/mL 7α,25-DHC for 48 h, respectively (Figure 6d). These data indicated that 7α,25-DHC induces inflammation through the upregulation of ROS and inflammatory mediators in L929 cells.

### 2.5. Crosstalk between p53 and Akt Cellular Signaling Pathway Mediates 7α,25-DHC-Induced Autophagy in L929 Cells

To evaluate whether 7α,25-DHC induced autophagy, cells were stained using an autophagy detection kit (ab139484, Abcam, Waltham, MA, USA). The number of cells decreased in L929 cells incubated with 7α,25-DHC, similar to previous results of live/dead cell staining, DAPI staining, and H&E staining (Figure 7a). In contrast, the number of L929 cells undergoing autophagy was increased by 7α,25-DHC. To evaluate the alterations of autophagy biomarkers such as beclin-1 and microtubule-associated protein 1A/1B-light chain 3 (LC3) by Western blotting, L929 cells were incubated with 1, 5, 10, or 25 μg/mL 7α,25-DHC for 48 h. The expression of beclin-1 (46 kDa), LC3-I (16 kDa), and LC3-II (14 kDa) were enhanced in L929 cells incubated with 7α,25-DHC (Figure 7b). These data indicate that 7α,25-DHC induces autophagy in L929 cells. To verify the cellular signaling pathways related to 7α,25-DHC-induced autophagy in L929 cells, the alteration of Akt and p53 were investigated in L929 cells incubated with 1, 5, 10, or 25 μg/mL 7α,25-DHC for 48 h. Our results revealed that phosphorylation of Akt was decreased in L929 cells incubated with 7α,25-DHC, whereas the expression of p53 was increased by 7α,25-DHC in L929 cells (Figure 7c). Our results suggest that 7α,25-DHC induces autophagy via crosstalk between p53 and Akt cellular signaling pathways in L929 cells.

### 2.6. GPR183 Regulates 7α,25-DHC-Induced Oxiapoptotic Cell Death in L929 Cells

To evaluate whether 7α,25-DHC-induced apoptosis, ROS production, and autophagy were regulated by GPR183, a receptor of 7α,25-DHC, cells were incubated with 25 μg/mL 7α,25-DHC in the presence or absence of 2.5 μM NIBR189, a GPR183 chemical inhibitor, for 48 h. Thereafter, alterations in cell viability, cell survival, apoptosis, ROS production, and autophagy were investigated, as shown in Figure 8. Cell viability was 93.02 ± 4.3% and 66.19 ± 2.8% (*p* < 0.01) in L929 cells incubated with 2.5 μM NIBR189 ((2E)-3-(4-Bromophenyl)-1-(4-(4-methoxybenzoyl)-1-piperazinyl)-2-propen-1-one; Sigma-Aldrich) and 25 μg/mL 7α,25-DHC, respectively, compared with control (100.02 ± 7.6%). Cell viability was measured as 80.21 ± 3.5% in L929 cells co-incubated with 2.5 μM NIBR189 and 25 μg/mL 7α,25-DHC (Figure 8a). Furthermore, similar to the results of cell viability, cell live/dead staining showed that NIBR189 counteracted 7α,25-DHC-induced cell death (Figure 8b). Hence, these data indicated that the inhibition of GPR-183 using NIBR189 counteracted 7α,25-DHC-induced cytotoxicity in L929 cells. The upregulation of caspase-9 and -3 in L929 cells was significantly suppressed by NIBR189 (Figure 8c). These data clearly indicate that 7α,25-DHC-induced apoptosis was suppressed by the inhibition of GPR183 by NIBR189. Moreover, NIBR189 not only suppressed ROS production (Figure 8d), but also inhibited the formation of autophagosomes and suppressed the expression of beclin-1 and LC-3 in L929 cells incubated with 7α,25-DHC (Figure 8e,f). These data consistently reveal that 7α,25-DHC-induced ROS production and autophagy are mediated by GPR183 in L929 cells. Our data suggest that 7α,25-DHC-induced oxiapoptotic cell death is regulated by GPR183 in L929 cells.

## 3. Discussion

Cholesterol, being a sterol, is not only an essential organic component that maintains the integrity and fluidity of mammalian cell membranes, but it is also involved in cell growth and proliferation through the biogenesis and functions of the cell membrane [26,27,28]. Furthermore, cholesterol not only serves as a precursor of bile acids, steroid hormones, and vitamin D [29], but it also acts as a ligand for various receptors, including nuclear receptors and GPCRs [30,31,32]. However, cholesterol, which is highly susceptible to oxidation, tends to form various oxysterol derivatives with polar groups such as hydroxy, keto, hydroperoxyl, epoxy, or carboxyl groups through either enzymatic activity, auto-oxidation, or both [33].

The enzymatic synthesis of oxysterol is carried out by oxidoreductases, hydrolases, and transferases. Oxysterols synthesized by these processes generally contain oxidized side chains [34]. Hence, genetic alterations in these enzymes are closely involved in carcinogenesis through abnormal synthesis of oxysterols from cholesterol [35,36]. Oxysterols synthesized by non-enzymatic processes associated with auto-oxidation of cholesterol have various types of molecules such as singlet oxygen, hydrogen peroxide, hydroxyl radicals, and ozone in their sterol rings. Furthermore, certain oxysterols, including 7-ketocholesterol (KC), 7α-HC, 7β-HC, and 25-HC, are synthesized via both enzymatic activity and ROS production pathway [37]. Although oxysterols are associated with the regulation of various biological processes, such as lipid and cholesterol metabolism, apoptosis, autophagy, and prenylation of proteins, their physiological action is still largely unknown.

Many studies reported that some oxysterols, including 25-HC, 5,6-epoxycholesterol isomers, 7β-HC, and 7-KC, induced oxiapoptophagy in various cell types [38,39,40,41]. Oxiapoptophagy is a cell death in which apoptosis, oxidative stress, and autophagy are combined. Previously, our studies reported the 25-HC-induced extrinsic and intrinsic apoptosis in primary rat chondrocytes and FaDu head and neck squamous cell carcinoma cells [10,42]. More recently, we reported 25-HC-induced oxiapoptophagy in L929 cells [39]. Therefore, as an extension of our previous study, we demonstrated that 7α,25-DHC, a downstream oxysterol synthesized from 25-HC by CYP7B1, induces oxiapoptophagy in L929 cells. In our previous study, 25-HC, an oxysterol upstream of 7α,25-DHC decreased cell viability and survival by increasing cell populations with condensed chromatin and altered morphology in L929 cells [39]. Similar to the results obtained for 25-HC, cytotoxicity was increased by 7α,25-DHC in L929 cells (Figure 2). Furthermore, DAPI and H&E staining showed typical apoptotic characteristics of L929 cells incubated with 7α,25-DHC (Figure 3). Moreover, FACS analysis revealed that the apoptotic population was increased by 7α,25-DHC exposure in L929 cells (Figure 4). Our study indicates that 7α,25-DHC induces apoptotic cell death in L929 cells.

Apoptosis is generally mediated by two apoptotic cellular signaling pathways: extrinsic and intrinsic [33]. The extrinsic pathway of apoptosis is initiated by the binding of death receptors with extracellular signaling molecules such as tumor necrosis factor (TNF), CD95-ligand (Fas-L), and TNF-related apoptosis-inducing ligand (TRAIL) [33]. Thereafter, death receptor-ligand binding induces the cleavage of inactive pro-caspase-8 into its active form. Subsequently, activated caspase-8 induces the activation of caspase-3 and PARP [33]. Intrinsic apoptosis is mediated by the reduction of mitochondrial membrane potential (ΔΨm), which results in the downregulation of anti-apoptotic molecules related to the maintenance of mitochondrial outer membrane integrity (such as Bcl-2 and Bcl-xL), and the upregulation of pro-apoptotic molecules (such as Bid, Bad, Bim, and Bax) related to the enhancement of mitochondrial outer membrane permeabilization [33,43]. Subsequently, cytochrome c released from the mitochondrial outer membrane with a decreasing ΔΨm induces the formation of an apoptosome composed of apoptotic protease-activating factor-1 (Apaf-1), pro-caspase-9, and ATP. Thereafter, cleaved caspase-9 induces cell death by sequential activation of caspase-3 and PARP, leading to oligonucleosomal DNA fragmentation [43].

As shown in Figure 5a, 7α,25-DHC decreased Bid expression, indicating that an increase in Bid, a mitochondrial membrane-targeted death ligand, induces Bak oligomerization [44]. 7α,25-DHC suppressed antiapoptotic molecules including Bcl-2 and Bcl-xL and upregulated pro-apoptotic molecules including Bax, Bad, and Bak. Consequently, activated caspase-9 was increased in L929 cells incubated with 7α,25-DHC. These data indicate that 7α,25-DHC released cytochrome c from the mitochondrial outer membrane into the cytosol through a decrease in ΔΨm in L929 cells. Finally, activated caspase-9 induced the activation of caspase-3 and PARP (Figure 5b,c). Collectively, these data indicate that 7α,25-DHC induces intrinsic apoptosis in L929 cells. However, intrinsic apoptosis is linked to mitochondrial dysfunction, which results in oxidative stress associated with ROS, including hydroxyl radicals (OH^−^), hydrogen peroxide (H_2_O_2_), and superoxide anions (O_2_^−^) [33]. Furthermore, ROS induces oxidative damage to mitochondrial lipids, DNA, and proteins [33]. Additionally, previous studies on oxysterol-induced cell death reported that ROS production is upregulated in various types of cells incubated with oxysterols including 5,6-epoxycholesterol isomers, 7-KC, and 7β-HC [40,41]. In our previous study, ROS production was significantly increased in L929 cells incubated with 25-HC [39]. Moreover, ROS production is closely associated with the progression of inflammation [45]. Similar to 25-HC, ROS production was enhanced by 7α,25-DHC in the L929 cells (Figure 6). Our results indicate that 7α,25-DHC-induced intrinsic apoptosis is involved in oxidative stress caused by enhanced ROS in L929 cells.

In the present study, autophagy staining showed that L929 cells with autophagic status were increased by 7α,25-DHC (Figure 7a). Furthermore, beclin-1 and LC3 expressions were significantly upregulated in L929 cells incubated with 7α,25-DHC (Figure 7b). Moreover, 7α,25-DHC suppressed Akt phosphorylation, whereas p53 phosphorylation was enhanced in L929 cells incubated with 7α,25-DHC (Figure 7c). The Akt cellular signaling pathway is closely associated with cell survival, proliferation, and growth during development and carcinogenesis through phosphorylation of forkhead box O (FOXO), a downstream target molecule of Akt [46]. Furthermore, FOXO mediates the role of p53, a tumor suppressor [46]. Hence, downregulation of Akt phosphorylation suppresses cell survival, proliferation, and growth through the upregulation of p53. p53 is a well-known transcription factor that regulates the expression of various pro-apoptotic genes, including cell cycle inhibitors, oxidative stress mediators, signaling molecules associated with death receptors, caspase activators, and the pro-apoptotic Bcl-2 family, which are related to mitochondrial outer membrane permeabilization [47]. 7α,25-DHC-induced apoptosis is involved in the Akt/p53 crosstalk axis in L929 cells. In addition, Akt/p53 crosstalk is closely associated with autophagy, known as autophagocytosis, which is a natural cellular degradation process that removes unnecessary or dysfunctional cellular components such as organelles, proteins, and lipids by autophagosome-containing lysosomes [33,39,48]. Generally, the phosphorylation of Akt, a vital messenger in the phosphoinositide 3-kinase (PI3K) pathway, induces mTOR Ser2448 phosphorylation to directly activate mTORC1, a master regulator of unc-51-like kinase 1 (ULK1), which is associated with the initiation of autophagy [47,49]. However, mTORC1 chemical inhibitors, such as rapamycin, upregulate ULK1 kinase activity [50]. In contrast, mTORC1 activation mediated by the Ras homolog enriched in the brain (Rheb) potently suppresses the kinase activity of ULK1 [51]. Therefore, Akt phosphorylation suppresses autophagy by suppressing ULK1 function to initiate the biogenesis of autophagosomes composed of beclin-1, LC3, and autophagy-related proteins (ATGs), including ATG5, ATG12, and ATG16L. Our data indicate that 7α,25-DHC-induced apoptosis is involved in autophagy via the Akt/p53 crosstalk axis in L929 cells. 

GPR183 is a receptor of 7α,25-DHC [52]. Sun and Liu have reported that the 7α,25-DHC-GPR183 axis contributes to inflammation and various diseases, including cancer, autoimmune diseases, cardiovascular diseases, and neurodegenerative diseases, through the regulation of various physiological properties such as migration, activation, and function in B cells, T cells, dendritic cells, and immune cells [13]. Furthermore, Bartlett et al. reported that the inhibition of GPR183 using GSK682753, a chemical antagonist of GPR183, decreased the expression of autophagy biomarker LC3 and autophagic flux in 7α,25-DHC stimulated Bacille de Calmette-Guerin vaccine (BCG)-infected THP-1 cells [22]. In this study, we showed that inhibition of GPR183 using NIBR189 significantly suppressed 7α,25-DHC-induced apoptosis, ROS production, and autophagy in L929 cells (Figure 8). Our results indicate that the 7α,25-DHC-GPR183 axis regulated oxiapoptophagy in L929 cells.

## 4. Materials and Methods

### 4.1. Cell Maintenance

L929 cells obtained from the American Type Culture Collection (Manassas, VA, USA) were maintained in Eagle’s minimum essential medium (EMEM) containing 10% fetal bovine serum (Welgene Inc., Gyeongsan, Korea) and 1% penicillin–streptomycin (Welgene Inc.) in a 5% CO_2_ incubator at 37 °C.

### 4.2. MTT Assay

The cytotoxicity of 7α,25-DHC was measured by MTT assay in L929 cells. L929 cells (1 × 10^5^ cells/mL) were cultured in 96-well plates for 24 h and then treated with 0, 1, 5, 10, and 25 μg/mL 7α,25-DHC for 48 h. MTT was added to L929 cells and incubated for 4 h. The resulting MTT crystals were resuspended in dimethyl sulfoxide (Sigma-Aldrich). Optical density was measured at 570 nm by a microplate spectrophotometer (Epoch; BioTek, Winooski, VT, USA).

### 4.3. Cell Live/Dead Staining

L929 cells (1 × 10^5^ cells/mL) were cultured in 8-well chamber slides (Nunc^®^ Lab-Tek^®^ Chamber Slide™ system, Sigma-Aldrich) for 24 h and then treated with 0, 1, 5, 10, and 25 μg/mL 7α,25-DHC for 48 h. Thereafter, cell survival was verified by a LIVE/DEAD™ cell assay kit (Thermo Fisher Scientific, Rockford, IL, USA) composed of calcein green AM to stain live cells with green fluorescence and ethidium homodimer-1 to stain dead cells with red fluorescence, according to the manufacturer’s instructions. Images were acquired under an inverted fluorescence microscope (Eclipse TE2000; Nikon Instruments, Inc., Melville, NY, USA). Histogram presented the relative % of total cells including live and dead cells. In addition, relative % of dead cells per total cells was calculated by (number of dead cells/(number of live cells + number of dead cells) × 100). Results were presented as mean ± standard deviation of three independent experiments, * *p* < 0.05 and ** *p* < 0.01.

### 4.4. DAPI Staining

Chromatin condensation was observed using DAPI staining to stain the nucleus of cells. Briefly, L929 cells (1 × 10^5^ cells/mL) were cultured in 8-well chamber slides for 24 h and then treated with 0, 1, 5, 10, or 25 μg/mL 7α,25-DHC for 48 h. After cultivation, the cells were rinsed with phosphate-buffered saline (PBS; Sigma-Aldrich), fixed with 4% paraformaldehyde, permeabilized with 0.1% triton X-100, and then stained with DAPI. Cells with condensed chromatin were taken under an inverted fluorescence microscope. Relative % of total cells was counted by cells with condensed chromatin and intact chromatin. Thereafter, relative % of cells with condensed chromatin was calculated by (number of cells with condensed chromatin/(number of cells with condensed chromatin + number of cells with intact chromatin) × 100). Results were presented as mean ± standard deviation of three independent experiments, * *p* < 0.05 and ** *p* < 0.01.

### 4.5. H&E Staining

Morphological alteration as cell shrinkage is a representative characteristic of apoptosis. Hence to observe the morphological alteration, L929 cells (1 × 10^5^ cells/mL) were cultured in 8-well chamber slides and allowed to adhere for 24 h. The cultured L929 cells were then treated with 0, 1, 5, 10, or 25 μg/mL 7α,25-DHC for 48 h at 37 °C. After cultivation, the cells were rinsed with PBS and fixed with 4% paraformaldehyde (Sigma-Aldrich) for 30 min at 4 °C. The morphology of L929 cells incubated with 7α,25-DHC was observed by H&E staining. Images of these cells were captured by a microscope (Leica DM750; Leica Microsystems, Heerbrugg, Switzerland). Relative % of total cells was counted by cells with both altered and intact morphologies. Thereafter, relative % of cells with altered morphologies was calculated by (number of cells with altered morphology/(number of cells with altered morphology + number of cells with intact morphology) × 100). Results were presented as mean ± standard deviation of three independent experiments, * *p* < 0.05 and ** *p* < 0.01.

### 4.6. Flow Cytometry Determination of Apoptosis

The apoptotic population was determined using FACS analysis. Briefly, L929 cells (1 × 10^5^ cells/mL) were cultured in 6-well culture plates and treated with 0, 1, 5, 10, and 25 μg/mL 7α,25-DHC for 48 h. Thereafter, the cells were rinsed with ice-cold PBS resuspended in binding buffer (BD Biosciences, San Diego, CA, USA), and then incubated with annexin V-fluorescein isothiocyanate (annexin V-FITC) and PI (Cell Signaling Technology, Danvers, MA, USA) for 15 min at 37 °C to perform FACS analysis using an automated high-speed cytometer sorting system (BD Biosciences).

### 4.7. Western Blotting Analysis

L929 cells (1 × 10^5^ cells/mL) were cultured in 6-well plates and treated with 0, 1, 5, or 10, and 25 μg/mL 7α,25-DHC for 48 h. The concentrations of total proteins extracted from the cells by cell lysis buffer (Cell Signaling Technology) were quantified using the bicinchoninic acid protein assay (Thermo Fisher Scientific). The same concentration of each protein sample was loaded onto 10% sodium dodecyl sulfate-polyacrylamide gel and then transferred to polyvinylidene fluoride (PVDF) membranes (Millipore, Burlington, MA, USA) at 4 °C. Subsequently, the PVDF membrane was blocked using 5% (*v*/*v*) bovine serum albumin (BSA; Sigma-Aldrich) in Tris-buffered saline with Tween-20 (TBS-T; Santa Cruz Biotechnology Inc., Dallas, TX, USA) and then incubated with primary antibodies at 4 °C for 12 h. Following, antibodies were purchased from Santa Cruz Biotechnology, Inc.: B-cell lymphoma-2 (Bcl-2; Cat. No. sc-7382, 1:1000), B-cell lymphoma-extra-large (Bcl-xL; Cat. No. sc-8392, 1:1000), β-actin (Cat. No. Sc-47778, 1:2000), p53 (Cat. No. sc-126, 1:1000), and beclin-1(Cat. No. sc-48341, 1:1000). The following antibodies were purchased from Cell Signaling Technology: cleaved-caspase-3 (Cat No. 9664T, 1:1000), caspase-9 (Cat. No. 9508S, 1:1000), Bcl-2-associated X protein (Bax; Cat. No. 2772, 1:1000), Bcl-2-associated agonist of cell death (Bad; Cat. No. 9239, 1:1000), Bcl-2 homologous antagonist killer (Bak; Cat. No. 12105, 1:1000), poly (ADP-ribose) polymerase (PARP; Cat. No. 9542S, 1:1000), cyclooxygenase-2 (COX-2; Cat. No. 12282, 1:1000), phospho-serine-threonine kinase (*p*-Akt; Cat. No. 4060, 1:1000), total Akt (Cat. No. 4685, 1:1000), and microtubule-associated protein 1A/1B-light chain (LC3; Cat. No. 12741, 1:1000). The membranes were rinsed three times with TBS-T and incubated with horseradish peroxidase (HRP)-conjugated secondary antibody (1:5000) for 2 h. Immunoreactive bands visualized using an ECL System (Amersham Biosciences, Piscataway, NJ, USA) were taken by MicroChemi 4.2 (Dong-Il Shimadzu Corp., Seoul, Korea). Thereafter, densitometric analysis was performed using ImageJ bundled with Java 1.8.0_172 software.

### 4.8. Caspase-3 Activity Assay

L929 cells (1 × 10^5^ cells/mL) were cultured in 8-well chamber slides and allowed to adhere for 48 h. Cultured L929 cells were treated with 0, 1, 5, 10, or 25 µg/mL 7α,25-DHC for 48 h at 37 °C. Thereafter, the activity of caspase-3 was assessed by the cell-permeable fluorogenic substrate PhiPhiLux-G_1_D_2_ (OncoImmunin Inc., Gaithersburg, MD, USA). The image of cells was taken by an inverted fluorescence microscope.

### 4.9. Cellular ROS Staining

After incubation under previously described cell culture conditions, ROS was detected by 5 μM DCFDA staining at 37 °C for 30 min. The image of cells was taken by a laser confocal scanning microscope (Leica Microsystems). ROS intensity was verified using an automated high-speed cytometer sorting system with BD Cell Quest version 3.3. Relative ROS production was measured using a fluorescence microplate reader (Bio Tek). Fluorescent was measured at an excitation wavelength of 485 nm and an emission wavelength of 520 nm.

### 4.10. PGE_2_ Assay

L929 cells (1 × 10^5^ cells/mL) cultured on 6-well plates were treated with 0, 1, 5, or 10, and 25 μg/mL 7α,25-DHC for 48 h. Thereafter, PGE_2_ production was measured by a Parameter™ PGE2 assay kit (Thermo Fisher Scientific) according to the manufacturer’s protocol.

### 4.11. Detection of Autophagy

L929 cells (1 × 10^5^ cells/mL) were cultured in 8-well chamber slides and allowed to adhere for 24 h. Cultured cells were treated with 0, 1, 5, 10, or 25 µg/mL 7α,25-DHC for 48 h at 37 °C. Thereafter, autophagy was detected using an autophagy detection kit (ab139484, Abcam, Waltham, MA, USA), according to the manufacturer’s protocol. The image of cells was taken using a laser confocal scanning microscope.

### 4.12. Statistical Analysis

Statistical differences were calculated using ANOVA and Tukey’s multiple comparison test. All data are presented as mean ± standard deviation. Statistical significance was set at *p* < 0.05.

## 5. Conclusions

This study demonstrated that 7α,25-DHC induced cell death through oxiapoptophagy accompanied with apoptosis, oxidative stress and autophagy by the crosstalk of Akt and p53 cellular signaling pathways in L929 fibroblast cells. Hence, our study suggests that the synthesis of 25-HC-7α,25-DHC from cholesterol by the enhancement of inflammation is closely associated with progressive degeneration of fibroblastic tissues through the induction of oxiapoptophagic cell death. Furthermore, our study suggests that the inhibition of GPR183, a receptor of 7α,25-DHC, might suppress or retard 7α,25-DHC-induced fibroblastic tissue degeneration.

## Figures and Tables

**Figure 1 molecules-27-04798-f001:**
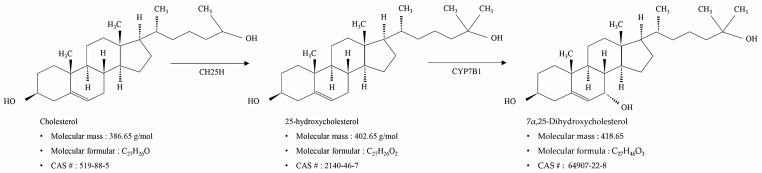
Synthesis and chemical information of 7α,25-dihydroxycholesterol (7α,25-DHC).

**Figure 2 molecules-27-04798-f002:**
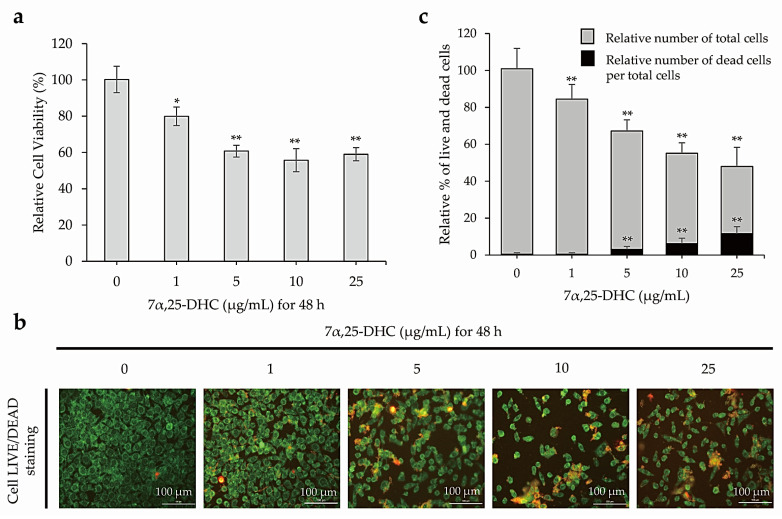
Cell death was increased in L929 cells treated with 7α,25-DHC. MTT assays and live/dead cell staining were performed to verify cytotoxicity and cell survival, respectively, in L929 cells incubated with 1, 5, 10, and 25 μg/mL 7α,25-DHC for 48 h. (**a**) Cytotoxicity was increased in L929 cells incubated with 7α,25-DHC. (**b**) 7α,25-DHC decreased the survival of L929 cells. Live cells were stained as green fluorescence by green calcein AM. Dead cells were stained as red fluorescence by ethidium homodimer-1. (**c**) Cell survival was decreased by 7α,25-DHC exposure of L929 cells. Histogram presents the relative % of total cells (gray color) including live and dead cells. In addition, relative % of dead (black color) cells per total cells was calculated by (number of dead cells/(number of live cells + number of dead cells) × 100). Results are mean ± standard deviation of three independent experiments, * *p* < 0.05 and ** *p* < 0.01.

**Figure 3 molecules-27-04798-f003:**
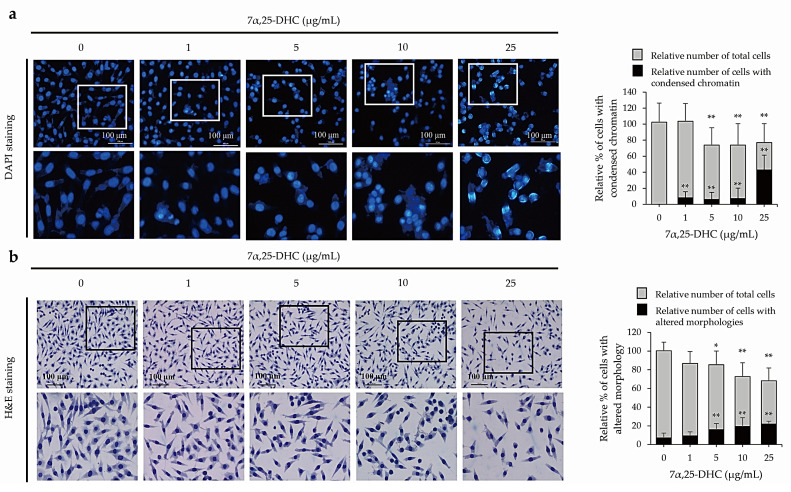
7α,25-DHC induced apoptotic cell death in L929 cells. To verify the apoptotic characteristics, L929 cells were incubated with 1, 5, 10, and 25 μg/mL 7α,25-DHC for 48 h. Thereafter, chromatin condensation and cell morphology were investigated by DAPI and H&E staining, respectively. (**a**) 7α,25-DHC enhanced the abundance of L929 cells with condensed chromatin. The nuclei of the cells were stained as blue fluorescence by DAPI. Relative % of total cells (gray color) were counted by cells with condensed chromatin and intact chromatin. Thereafter, relative % of cells with condensed chromatin (black color) was calculated by (number of cells with condensed chromatin/(number of cells with condensed chromatin + number of cells with intact chromatin) × 100). (**b**) Cells with an altered morphology were increased in L929 cells incubated with 7α,25-DHC. Relative % of total cells (gray color) was counted by cells with both altered and intact morphologies. Thereafter, relative % of cells with altered morphologies was calculated by (number of cells with altered morphology/(number of cells with altered morphology + number of cells with intact morphology) × 100). Results are mean ± standard deviation of three independent experiments, * *p* < 0.05 and ** *p* < 0.01.

**Figure 4 molecules-27-04798-f004:**
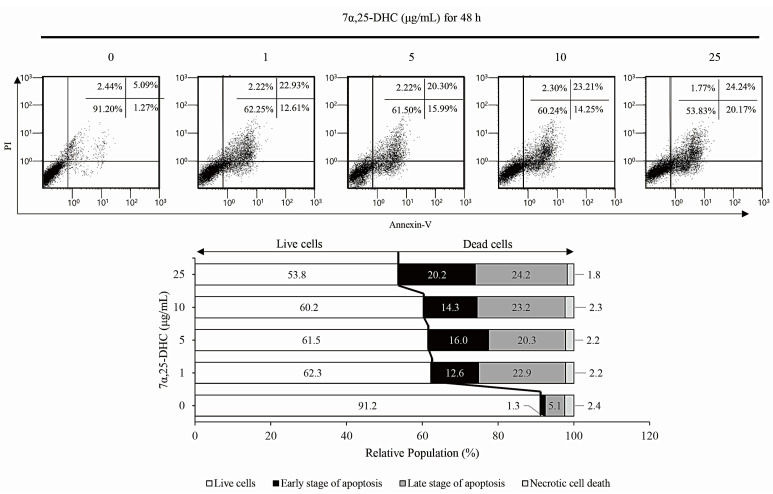
The apoptotic population was enhanced by 7α,25-DHC in L929 cells. Apoptotic population was analyzed by FACS using PI and annexin V labeling in L929 cells incubated with 1, 5, 10, and 25 μg/mL 7α,25-DHC for 48 h.

**Figure 5 molecules-27-04798-f005:**
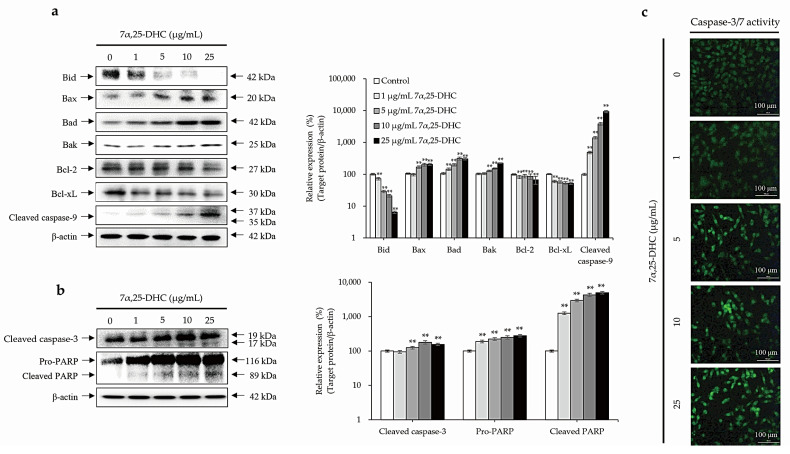
7α,25-DHC-induced cell death was involved with the intrinsic apoptosis pathway in L929 cells. L929 cells were incubated with 1, 5, 10, and 25 μg/mL 7α,25-DHC for 48 h. Total proteins were extracted using cell lysis buffer and then quantified by BCA assay kit. Sequentially, equal concentrations of protein samples were loaded onto SDS-PAGE gels to perform Western blotting using antibodies against pro- and antiapoptotic molecules. Thereafter, immunoreactive bands were exposed by ECL solution and acquired under microChemi 4.2. Results are presented as the relative ratio of target protein/β-actin, ** *p* < 0.01. β-actin was used as an internal control. (**a**) 7α,25-DHC induced mitochondria-dependent intrinsic apoptosis in L929 cells. (**b**) Cleaved caspase-9 induced cell death through the activation of caspase-3 and PARP in the L929 cells treated with 7α,25-DHC. (**c**) The activity of caspase-3 was increased by 7α,25-DHC in L929 cells. Cells with activated caspase-3 were stained as green fluorescence by the cell-permeable fluorogenic substrate PhiPhiLux-G_1_D_2_.

**Figure 6 molecules-27-04798-f006:**
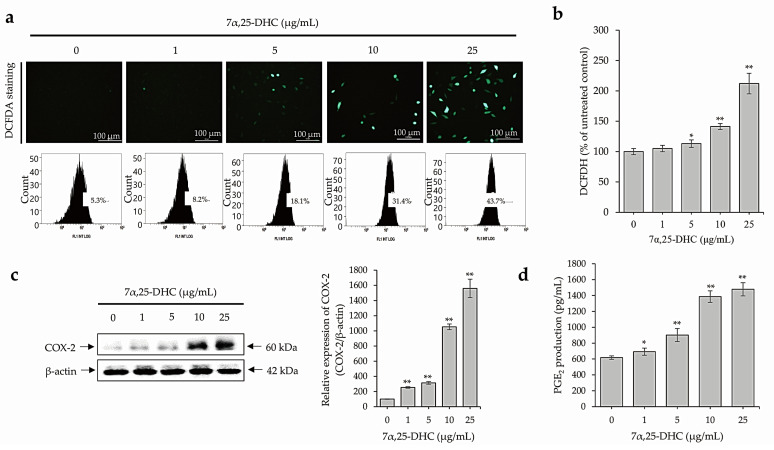
7α,25-DHC enhanced the production of ROS, the expression of COX-2, and the production of PGE_2_ in L929 cells. L929 cells incubated with 1, 5, 10, and 25 μg/mL 7α,25-DHC for 48 h. 2′,7′–dichlorofluorescein diacetate (DCFDA) staining (**a**) and intensity measurements using microplate reader (**b**) were performed to verify the production of ROS. In addition, Western blot using COX-2 antibody and PGE_2_ assay were performed to verify the ROS-mediated upregulation of inflammatory mediators in L929 cells treated with 7α,25-DHC. (**a**) 7α,25-DHC increased the ROS production in L929 cells. ROS produced by cells treated with 7α,25-DHC were stained as green fluorescence by DCFDA. In addition, cells stained by DCFDA were counted by FACS. (**b**) ROS intensity was increased by 7α,25-DHC exposure in L929 cells. (**c**) The expression of COX-2 was increased in the L929 cells treated with 7α,25-DHC. (**d**) The production of PGE_2_, a downstream inflammatory mediator of COX-2, was upregulated by 7α,25-DHC in L929 cells. Results are mean ± standard deviation of three independent experiments, * *p* < 0.05 and ** *p* < 0.01.

**Figure 7 molecules-27-04798-f007:**
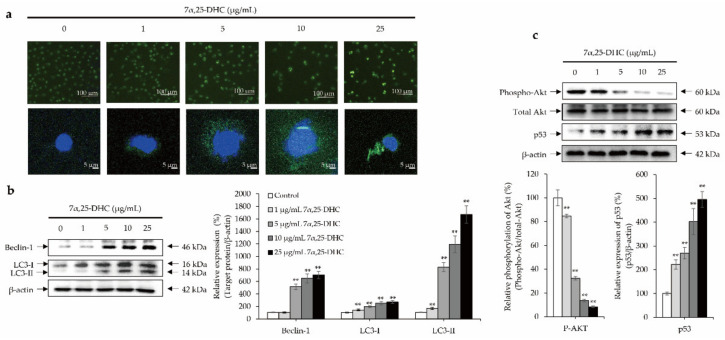
7α,25-DHC induced autophagy through Akt/p53 crosstalk in L929 cells. L929 cells were incubated with 1, 5, 10, or 25 μg/mL 7α,25-DHC for 48 h. Thereafter, autophagy staining was performed by an autophagy detection kit. Images were taken by a laser confocal scanning microscope. In addition, Total proteins were extracted using cell lysis buffer and then quantified by BCA assay kit. Sequentially, equal concentration of protein samples was loaded onto SDS-PAGE gels to perform Western blotting using beclin-1 and LC3 antibodies. Immunoreactive bands were exposed by ECL solution and acquired under microChemi 4.2. Results were presented as the relative ratio of target protein/β-actin, ** *p* < 0.01. β-actin was used as an internal control. (**a**) Autophagy-positive cell numbers were increased by 7α,25-DHC in L929 cells. Autophagic vacuoles and nucleus of cells were stained as green and blue fluorescence, respectively, by a cationic amphiphilic tracer dye and DAPI in L929 cells treated with 7α,25-DHC. (**b**) 7α,25-DHC increased the expression of beclin-1 and LC3 in L929 cells. (**c**) 7α,25-DHC induced Akt/p53 crosstalk in L929 cells.

**Figure 8 molecules-27-04798-f008:**
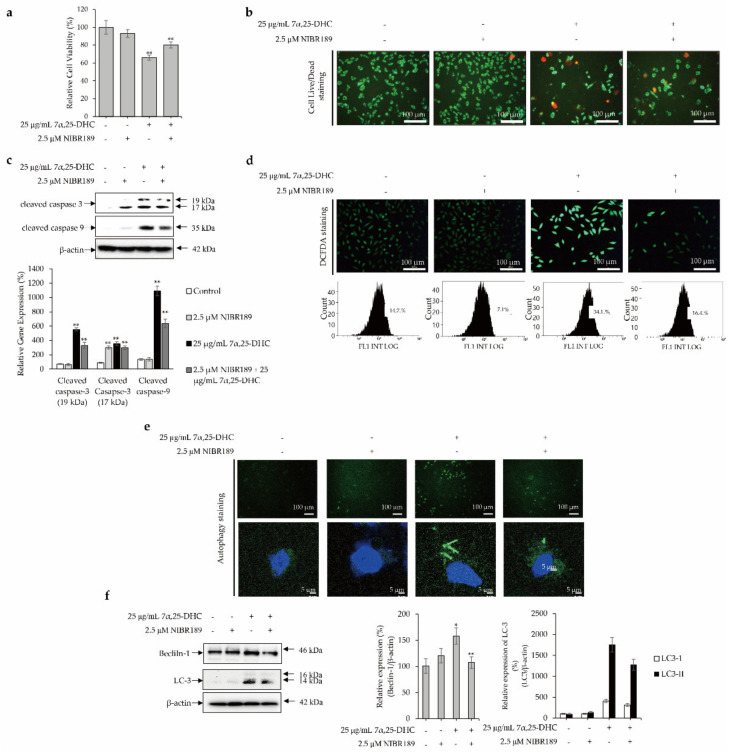
GPR183 regulates 7α,25-DHC-induced oxiapoptotic cell death in L929 cells. L929 cells were incubated with 25 μg/mL 7α,25-DHC in the presence or absence of 2.5 μM NIBR189 (chemical inhibitor of GP183) for 48 h. Results of western blot were presented as the relative ratio of target protein/β-actin, * *p* < 0.05 and ** *p* < 0.01. β-actin was used as an internal control. (**a**) NIBR189 counteracted 7α,25-DHC-induced cytotoxicity in L929 cells. (**b**) 7α,25-DHC-induced cell death was suppressed by NIBR189. Dead and live cells were stained as red and green fluoresces, respectively, by green calcein AM and ethidium homodimer-1. (**c**) NIBR189 downregulated the expression of caspase-9 and caspase-3 in L929 cells incubated with 7α,25-DHC. (**d**) NIBR189 suppressed the 7α,25-DHC-induced ROS production in L929 cells. ROS produced by cells treated with 7α,25-DHC were stained as green fluorescence by DCFDA. (**e**) Formation of autophagosome was reduced by NIBR189 in L929 cells incubated with 7α,25-DHC. Autophagic vacuoles and nucleus of cells were stained as green and blue fluorescence, respectively, by a cationic amphiphilic tracer dye and DAPI in L929 cells treat-ed with 7α,25-DHC. (**f**) The expression of autophagy biomarkers such as beclin-1 and LC3 were decreased by NIBR189 in L929 cells incubated with 7α,25-DHC.

## Data Availability

The data presented in this study are available on request from the corresponding author.

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
