# Peer review of "GPR183 Regulates 7α,25-Dihydroxycholesterol-Induced Oxiapoptophagy in L929 Mouse Fibroblast Cell"

_molecules, 2022, doi:10.3390/molecules27154798_

Round 1

Reviewer 1 Report

Kim and colleagues investigated the apoptosis, oxidative stress, and autophagy function of 7α,25-Dihydroxycholesterol in the Mouse Fibroblast Cell line. They have found that NIBR189, as an antagonist of GPR183 improved the 7α,25-DHC-induced cell death. However, some clarifications would be helpful.

Comments:

1.       Why did you choose L929 Mouse Fibroblast Cell. Did you know the metabolic concentration of 7α,25-DHC in plasma or serum in normal and pathology conditions? Did the level injure the cells? Please claim more clearly the value of clinical application for this research in the introduction section.

2.       The differences in quantification shown in figure 2c for the live and dead sections do not accurately reflect in the representative images shown in figure 2b, especially for 10ug/ml and25 ug/ml. These need to be changed to be able to correlate to the readers. Individual dots should be added to the bar graph to make the figure more reliable.

3.       Which reagent you use to dissolve the 7α,25-DHC, there is no description in the method section. Some of the reagents like DMSO will affect cell death as well, did you have the treatment group like accordingly amount of reagent for cells to measure apoptosis and autophagy etc.?

4.       For apoptotic phenotypes, DAPI should be used in fixed cells because of the low cell membrane-permeant and toxicity. Hoechst dyes were suggested to be used in the live cell, please check the protocol and method. H&E staining showed the morphology, but no description was found in the method and result section, it’s better to add them to give the reader a better understanding.  Zoom and high magnification representative images should be proved in this figure as well.

5.       Another suggestion is that make the legend clear, in terms lot of images in the figure, the color of the marker should be explicitly labeled in the legend or figure by color, one example is that in figure 8, serval green and blue and dismissed the label. The authors also need to state the number of cells or fluorescence density in figures 8b,d, and e All these details need to be mentioned so the readers can easily reach out.

6.       All the experiments used 48 hours and ensure they are all mentioned in the result section (missed in MTT assay, etc.).

7.       As the key hypothesis that 7α,25-DHC induces OXIAPOPTOPHAGY, the authors need to perform one or two additional oxidative stress function assays like SOD activity, catalase activity, GPx activity, lipid peroxidation ect. since the GPR183 could improve it

Overall there needs to be a significant improvement in the presentation of the results.

Author Response

Thank you for your encouragement to resubmit a revised version of our manuscript entitled, “GPR183 regulates 7α,25-dihydroxycholesterol-induced oxiapoptophagy in L929 mouse fibroblast cell”. We thank you and the reviewers for giving us the opportunity to revise our manuscript and for providing valuable comments and suggestions. In response to the reviewers’ comments, we have revised our original manuscript to improve the quality of our findings and strengthen our original conclusions. Please see attached "Response to reviewer #1"

Reviewer 2 Report

The manuscript completes the article also published in MOLECULES, last December - deepening the control mechanisms of oxiapoptophagy of fibroblasts. It is well explained, contains accurate data and reproducible experiments. The article can be published in the form in which it is.

Author Response

Thank you for your encouragement to resubmit a revised version of our manuscript entitled, “GPR183 regulates 7α,25-dihydroxycholesterol-induced oxiapoptophagy in L929 mouse fibroblast cell”. We thank you and the reviewers for giving us the opportunity to revise our manuscript and for providing valuable comments and suggestions. In response to the reviewers’ comments, we have revised our original manuscript to improve the quality of our findings and strengthen our original conclusions.

Reviewer 3 Report

The general work is novel. It has been described that 7α,25-Dihydroxycholesterol induces apoptosis and accumulation of cholesterol in other cell types (such as mouse and human liver cells). However, the effect of GPR183 on fibroblasts was not known. In addition, they evaluated different processes of cells in apoptosis in various stages.

However, some minor details needed to be improved to make the text more harmonious and the results easier to read.

Major comments

1.- The figure description should change to something more descriptive. That means explaining what is seen in the figures and not explaining the results. Give detail about what statistics studies were used in the different analyses, how many samples were used, the objective used in the microscope images, the dilution of the antibodies, the incubation times, etc. With this, it is easy to read the figure without going into the methodology. Please change that in all the figures.

2.- In figure 3b, if you have some photos with higher magnification to be able to see what they mean, since being so small, you can only see that the proliferation decreased, but you cannot notice a change in morphology.

-In this figure, figure 3a, why did you evaluate chromatin condensation using dapi? It is much clearer to assess it using FACS; since they did other FACS analyses, you could have considered it. Or have evaluated at least DNA fragmentation by TUNEL.

3.- You must improve the methodology. A lot of information is missing in all the techniques; please add the information about the antibodies (dilution, concentration, company where they are from, which secondary antibodies were used). The objective used in the microscope images was to take the photos. For ROS, what concentration, incubation time, and temperature were DCFDA used. How many n (sample number) was used in the WB and microscopy? These figures are representative of how many repetitions?. Please complete all the information.

3.- The conclusion is very broad, it should be more associated with what was discovered in this work and how this affects cell death. It can improve.

minor comments

1.- in figure 1, please keep the same format, molecular mass or weight?, # or No. Will be used, always in the same order, molecular mass, molecular formula, CAS. Or in the order, you choose but always the same g/mol to the last molecule.

2.- In figure 2, use the same scale on the Y axis, up to 120%

3.- Line 172, change belcin-1 to beclin-1

4.- Line 334, change CO2 to CO2

5.- Line 280, change H2O2 to H2O2

6.- Line 280, change OH- to OH-

7.- Line 290, change belcin-1 to beclin-1

8.- Line 298, change P53 to p53

9.- Line 303, change ATK/p53 to AKT/p53

10.- Line 27, change belcin-1 to beclin-1

Author Response

Thank you for your encouragement to resubmit a revised version of our manuscript entitled, “GPR183 regulates 7α,25-dihydroxycholesterol-induced oxiapoptophagy in L929 mouse fibroblast cell”. We thank you and the reviewers for giving us the opportunity to revise our manuscript and for providing valuable comments and suggestions. In response to the reviewers’ comments, we have revised our original manuscript to improve the quality of our findings and strengthen our original conclusions. Please see attached "Response to reviewer #3"

Round 2

Reviewer 1 Report

Agree to accept.